# Prevalence patterns of overweight and obesity in the world: An age-period-cohort analysis

**Ali Kabir** [1,2]*, **Arman Karimi Behnagh**[1]

**1** Minimally Invasive Surgery Research Center, Iran University of Medical Sciences, Tehran, Iran,
**2** Artificial Intelligence in Health Research Center, Iran University of Medical Sciences, Tehran, Iran

* kabir.a@iums.ac.ir, aikabir@yahoo.com

## Abstract

### Purpose

This study aimed to evaluate the trends of obesity and overweight among adolescents using the age-period-cohort (APC) analysis.

### Methods

Data for this study was provided by Institute for Health Metrics and Evaluation. Our data represents the cumulative prevalence trend of obesity and overweight in 195 countries between 1980 and 2015 in 5-year intervals. The age intervals were also considered to be 5 years. Besides, a subgroup analysis based on sex and socio-demographic-index subgroups were performed. To perform APC analysis, R program software was used.

### Results

We observed an increasing trend in both obesity and overweight throughout the study period, with the trend accelerating in more recent periods. Among the fitted models, we concluded that the APC model best fit the current data. The trend for both outcomes was similar. Among the three parameters, age showed an inverted U-shaped effect on the trend of both outcomes in all subgroups. However, the effect of period and cohort differed in our subgroup analysis.

### Conclusion

Overall, this study shows that obesity and overweight are on the rise. Both phenomena were influenced by the age effect in a similar pattern. However, the period and cohort effects showed variation in our subgroup analyses based on sex and SDI subgroups, suggesting the need for country-level studies to better understand the possible impact of these two factors on the prevalence of obesity and overweight.

**Data availability statement:** All the data regarding number of obese and overweight people is freely available from: https://ghdx.healthdata.org/record/ihme-data/gbd-2015-obesity-and-overweight-prevalence-1980-2015

**Funding:** This work was supported by National Institute for Medical Research Development (NIMAD) (Grant numbers [971515]). Dr. Ali Kabir has received this research grant. The funders had no role in study design, data collection and analysis, decision to publish, or preparation of the manuscript

## Introduction

In recent decades, the epidemic of obesity and overweight has become a global point of concern for health care systems, especially due to its influence on chronic conditions and overall life expectancy [1,2]. Owing to a combination of environmental factors, such as economic growth, sedentary lifestyle and increased intake of processed and ultra-processed food, the prevalence of obesity has been climbing steeply [1,2]. According to the latest World Health organization (WHO) report on obesity and overweight, the population of obese and overweight increased more than double since 1990 and currently almost 2.5 billion adult people live with obesity or overweight [3].

Based on the WHO reports, currently the obesity as defined as individuals who had body mass index equal or greater than 30 Kg/m². Similarly, the overweigh person has been defines as an adult who has BMI between 25 and 29.9 Kg/m² [3]. It should be noted that these cut-offs may not be applicable for all age groups and there are other cut-offs for adolescent, children and infants with obesity or overweight [3]. Furthermore, in some racial adjustments for these cut-offs in different geographic region.

Comparing to normal population, obese and overweight populations are at higher risk of poor health outcome during their life course [4,5]. Excess body weight is associated with adverse health outcomes on longevity, disability-free life years, quality of life, and productivity [4,5]. Moreover, a growing prevalence of childhood obesity, in particular, predicts an overwhelming burden of disease in the decades to come since there are close correlation between the childhood obesity or overweight with being obese later in life [6,7]. However, such patterns can be predicted and also the high-risk groups can be determined using particular statistical methods. Very recently, there have been reports on the trend of global obesity globally. The results of these studies further support the finding of WHO report [5,8].

Age period cohort (APC) analysis is a statistical approach by which time-varying variables may be better understood. In the field of epidemiology, these analyses are employed to distinguish the effects of three types of variables: age, period, and cohort [9]. Age effects are the result of the intertwined social and biological factors which are often specific to each individual population. It must be noted that age effects arise from physiological alterations and social experiences only related to ageing. These effects are usually characterized by the different disease profiles observed in each age group [10]. The structured of world population demographics has been changed significantly, as the fertility rates dropped considerably in comparison to the previous decades and simultaneously the life expectancy has been increased in most part the globe [11,12]. This issue shifts the age distribution of the world leading to an inflated rate of senile population globally. Previously it was shown that this could be associated with increased risk of obesity [13].

It's crucial to understand that any modifications in the study methodology—whether related to definitions, data collection, or classification—can manifest as period effects [14]. A period effect refers to the shifts that occur across different time frames, impacting individuals of all ages and backgrounds simultaneously. Additionally, recent developments in agricultural practices [15], advances in technologies [15],

climate changes [16], and increase in the food availability [17] lead increase the risk of obesity. These changes can also count as the period effect.

The final parameter of interest in APCs is the cohort effect, which are caused by the unique event exposures of population subgroups throughout time. Birth cohort is the most common among the cohort effects and is defined as the differences in outcome risks based on the year of birth [18]. Cohort effects reflect the differences in distribution between age groups when faced with the same exposure. To simplify, cohort effects are experienced differently by each age group due to the result of age-specific features or susceptibility to an exposure or event [18]. It is believed that younger generations may be more susceptible to an environment that promotes obesity, putting them at an increased risk for weight gain. This vulnerability is attributed to their early and widespread embrace of a modern lifestyle that involves a screen-oriented, sedentary routine coupled with a diet rich in calories [19].

To date there have been several studies in which the effect of each of these time-varying factors have been estimated [20,21]. In a report from US, it was shown inverted U-shaped age effect, meaning obesity risk increases to midlife and then declines. There's a strong period effect, with obesity prevalence rising over time across all groups. No significant cohort effect was found after adjusting for age and period effects. This suggests that nationwide environmental changes, rather than generational differences, drive the obesity epidemic [20]. Schramm S. et al. showed that the increase in the prevalence of obesity in Denmark was consistent across all age groups and sexes. The age effect peaked at 55–64 years. Period effects were significant, with each subsequent survey year showing higher obesity rates [21].

Majority of these studies focused on the regional and national trend of obesity and not the overweight. Furthermore, although picturing an overall view over all nations cannot be a suitable solution, such picture can disclose the similarities and common features that can be addressed and resolved by global policy makers. In this respect, we aimed to address the effect of each parameter in trend of overweight and obesity worldwide and different regions.

## Method and materials

### Data source and description

Data was provided by the IHME, based on a study of GBD series [2] and freely available in the this repository: https://ghdx.healthdata.org/record/ihme-data/gbd-2015-obesity-and-overweight-prevalence-1980-2015. Despite improvements in quality of the data modelled and produced by IHME, there are several reasons that made us use an older version of GBD studies' datasets and that is the fact that in the recent databases, the obesity and overweight data were provided in a combined manner and under title of "high BMI" or in a very new version of the published dataset, the data was provided in a rate format not the raw numbers [22].

Moreover, the study period introduced in the dataset can be a problem for addressing the obesity as a global public health issue. However, using such a dataset can be a solution for possible problems when it comes to the APC analysis. First of all, it is necessary to use a longitudinal data to capture the trend of the disease, here obesity. Then it is of vital importance to use a dataset in which data was collected based on a robust, valid and coherence methodologies. This issue is of vital importance, since using different datasets would increase the presence of methodologic biases. Therefore, there was no other option for use to use the a GBD dataset.

According to the reference study [2], systematic searches in Medline and the Global Health Data Exchange were conducted for nationally representative BMI data. In the reference study the following cut-offs, $\geq 25\,kg/m^2$ and $\geq 30\,kg/m^2$ were used to define overweight and obesity, respectively. After data gathering, a mixed effects model was used to approximate and correct the data of each GBD region and age group due to particular biases like the self-report [2]. Moreover, spatiotemporal Gaussian process regression was used to obtain the mean prevalences of overweight and obesity. In case of any missing data existence, the data was calculated from parameters and covariates of neighboring countries data using a linear model.

Based on the above-mentioned explanations, we used the data of 195 countries to estimate the global prevalence of obesity and overweight. The mean number and rate of obese and overweight population based on different age and sex groups was used for our analysis. As pre-requisite of APC analysis, the age and period intervals were set to be 5-year. The data on world population was obtained from same source (https://ghdx.healthdata.org/record/ihme-data/gbd-2015-population-estimates-1970-2015). Due to the structure of our dataset for obesity and overweight, we restructured data of world population in a 5 years interval for periods and age groups. A similar approach was employed for gender subgroups.

It should be noted that the present study is in agreement with the Declaration of Helsinki. The Research Ethics Committees of National Institute for Medical Research Development (NIMAD) approved the study protocol with ethics code: IR.NIMAD.REC.1397.015 at 1 July 2018.

## The challenges of APC analysis

Our data on obesity and overweight prevalence is comprised of data on prevalence rate in a consecutive number of years. To run an APC analysis, the model below is employed as a standard to discriminate the effects of age, period, and cohort from each other:

$$\mu = \alpha age + \beta period + \gamma cohort + \delta \tag{1}$$

Since each parameter of in this equation can be determined if the two other parameters are available. This rises a considerable collinearity between the parameters, which make it hard to the specify the exclusive effect of each parameter [23]. This problem is called as identification problem in APC analysis, and so far, many solutions have been proposed to solve this problem [24]. As such solutions, constrained coefficients GLIM estimator, proxy variables approach, nonlinear parametric transformation approach, log-linear models, and intrinsic estimator method can be mentioned [9,18,24–26].

As a result of linear dependency between the three components provided in Equation 1, the model lacks ability to of identify the effect of each parameter in its current form. By knowing two out of three components, the other component of the equation can also be determined. As an instance, if one knows the value of age and period the third parameter, cohort, can subtracting the value of age from the value of period. Thus, this allows for an infinite solution pool, it becomes impossible to produce a correct estimate for the individual effect of each of the parameters. As a result, the unique assignment of the observed trends to each component becomes impossible.

Such an approach is faced with two inherent complications. First, strong prior information is crucial to the assumptions underlying the imposed constraints and empirical validation of these assumptions is not possible. Second, the results of the analyses are heavily influenced by these constraints, to a degree that different age, period, and cohort estimates may be estimated based on the choice of identified constraints [27]. To date, numerous approaches have tried to address the identification conundrum in APC analysis. The estimable function, hierarchical-model approach and constrained generalized linear model are examples of such endeavors, however, they still rely heavily on the imposed constraints and assumptions [27].

In this study, we have employed the method introduced by Kuang et al. [26] and Nielsen et al. [28]. Using double difference (2) of features of the model gave us the possibility of calculating the acceleration or growth rate of the trend rather than the effect itself. Moreover, using the method explained by Kaung et al. [26] it is possible to calculate both linear and non-linear effect of each time-varying parameters. The model is structured using a freely varying vector composed of three starting points, functions involving the unknown first differences, and the full set of second-order differences in the time effects. Kuang et al. considered a parameter (i.e., here prevalence) shaped by the second differences and the three predictor entries:

$$\Delta^2 \alpha_i = \Delta a_i - \Delta \alpha_{i-1} \tag{2}$$

$$\xi = (\mu 11,\ \mu 21,\ \mu 12,\ \Delta 2\,\alpha 3...,\ \Delta 2\,\alpha I,\ \Delta 2\,\beta 3...,\ \Delta 2\,\beta J,\ \Delta 2\,\gamma h1 + 3...,\ \Delta 2\,\gamma K - h2) \tag{3}$$

In this Equation 3, I and J represent indices of age, period. As a result, the contribution of age, cohort and period is modeled with regards to the cumulative sums of the double differences with respect to age, period and cohort, which are identifiable (Equation 4).

$$\mu_{ij} = \mu_{11} + (i-1)(\mu_{21}-\mu_{11}) + (j-1)(\mu_{12}-\mu_{11}) + \sum_{t=3}^{i}\sum_{s=3}^{t}\Delta^2\alpha_s + \sum_{t=3}^{j}\sum_{s=3}^{t}\Delta^2\beta_s + \sum_{t=3}^{i+j-1}\sum_{s=3}^{t}\Delta^2\gamma_s \tag{4}$$

A full description of the statistical method used here in this study can be found in the study of Kaung et al. [26]. Further considerations and modification that we applied for conduction of an overdispersed Poisson model can be found in B. Nielsen et al. study [29].

### Socio-demographic index

The Socio-demographic Index (SDI) combines indicators of a nation's economic status, education levels, and fertility trends to create a standardized measure of global socioeconomic development. In this research, the index was employed to classify 195 countries into distinct subgroups and to account for the influence of economic, educational, and fertility factors when analyzing the impacts of age, period, and cohort variables. The list of countries in each of the SDI categories is provided in S1 Table in S1 File and the country data was obtained from the primary GBD study [2].

### Statistical analysis

We estimated obesity and overweight prevalence rate for each five-year period from 1980 to 2015. To construct the rate of obesity and overweight prevalence, we used the world population data in a similar structure as the obesity and overweight prevalence. To perform APC analysis, we used the prevalence of obese and overweight individuals which plotted separately by age at report time, year of report, and birth cohort. Akaike Information Criterion (AIC) was used to assess the model fitness and smaller values indicate better fit. Deviance was calculated using two parameters, degree of freedom and log-likelihood. The following formula was considered: Deviance $= -2 \times$ the difference in log-likelihood values for the nested models. In order to evaluate the impact of different time periods and cohorts, we compared the performance of models that did not include these specific terms (such as models with only one time parameter, PC, AC, and AP) with the performance of the model that included all three terms, APC. For the purpose of this study, we calculate the prevalence rate of obesity and overweight in each period, by dividing the number of people of with BMI > 30 (for obesity) and $25 \geq$ BMI > 30 (for overweight) to number of people in the specific period. For APC analysis usually the Poisson link function has been used as the data is count and can follow discrete Poisson distribution. However, in this work we applied a very specific approach to choose the best link function. Initially, we fit a Poisson model to the data to check for the goodness of fit. In majority of cases we realized that the division of deviance to the degree of freedom is greater than 1, which translates to the overdispersion in a Poisson model [29]. Therefore, we applied a quasi-Poisson model to the data for estimating the time effect. The model assumes that the observations are independent. Moreover, we applied the sensitivity analysis by using a Gaussian link function (while considering log normal distribution over data) as the datasets are cumulative distribution of multiple discrete Poisson distribution. The R package "apc" (version 2) was used to performed all the statistical analysis. All the analysis was performed in R (version 4.2.1, R core team, 2022).

### Results

Obesity is on the rise worldwide. Prevalence increased with age until the Middle Ages (around 50s and 60s age groups) and then declined however there is no consistent pattern on the peak respecting to the age groups. Recent periods had

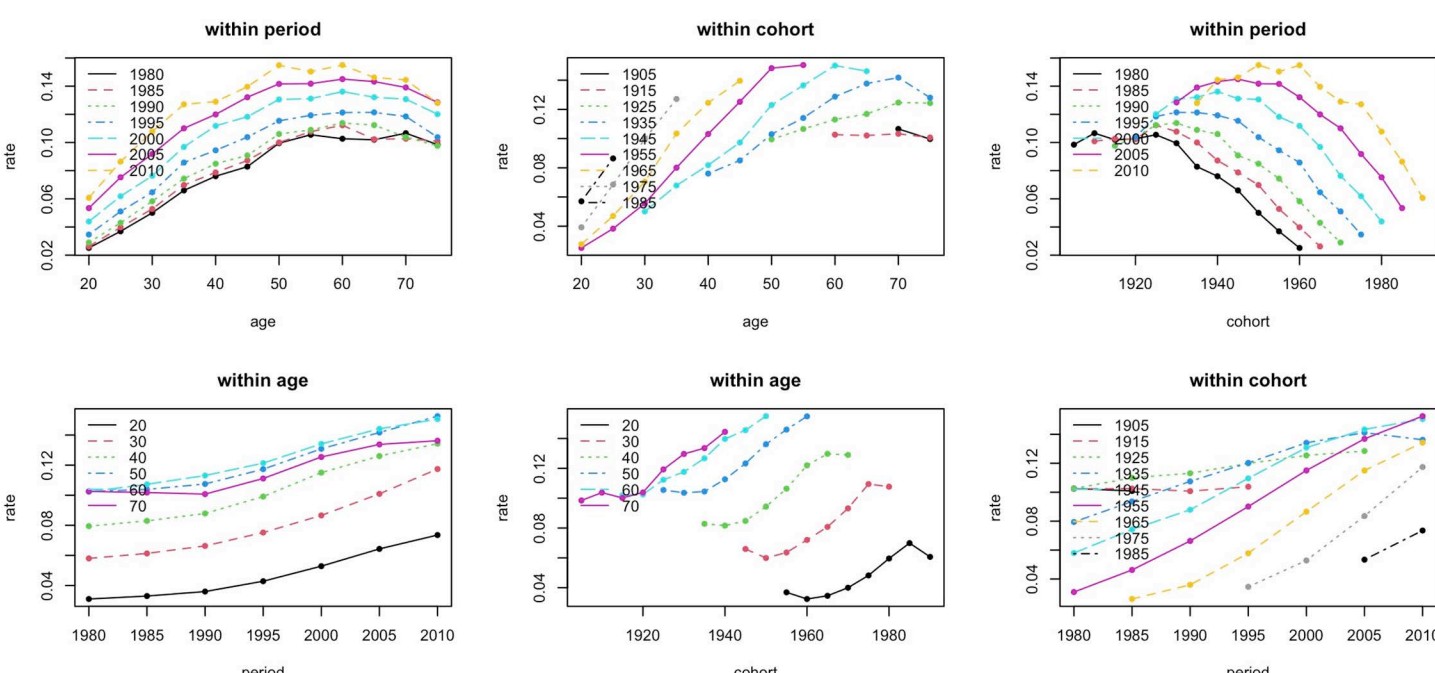

**Fig 1. The trend of the obesity prevalence rate within period-age, cohort-age and period-cohort for both sexes.** It should be noted that in these plots, each age and period group represent an interval of 5 years; for example, the period 1980 would be converted to 1980-1984. Based on the **upper left plot**, regardless of the time period, the age distribution of obesity prevalence has similar trend patterns. The **top middle plot** shows the trend of obesity prevalence in different cohort groups as they age through the time course. The **upper right plot** shows the trend of obesity prevalence for different cohorts during each period. Based on this plot, a vertical shift can be seen in the increasing periods. However, there is no significant difference in the shape of the trend line for each period. The **lower left plot** shows the trend for each age group as the period increases. Based on this plot, each age group had a higher prevalence of obesity compared to the previous period. The **lower middle plot** shows the trend for the same age groups of different cohorts. For example, the trend of the age group 20, born around 1970, had a lower prevalence compared to the 20s, born around 1980. Finally, the **lower left plot** shows the change in the trend of obesity prevalence for each cohort group over time. It can be seen that for the majority of cohorts, with the exception of the 1905 and 1915 cohorts, the rate increased after each period.

higher rate of obese populations, however, the age distribution of obesity over different periods had slight changes (Fig 1). According to Fig 1, it can be seen that the rate of obesity is the highest in age of 50s in the most recent period (i.e., 2010–2014). A similar pattern can be seen for population of high SDI countries (S1 Fig in S1 File). However, it should be noted that the peak prevalence rate was higher in high SDI countries in comparison to global rate (24% vs 15%, respectively). The pattern of obesity prevalence trend in low SDI countries was much more homogenous and the peak age of obesity rate was seen among the people at age group of 60–64 with rate of around 9% (S2 Fig in S1 File). The pattern of obesity in the rest of the SDI subgroups were represented in S3–S5 Figs in S1 File.

Our subgroup analysis based on sex, revealed different variations in the patterns of obesity trend in males. As an instance, in the recent period 2005–2009 and 2010–2014, there was a sudden increase in the rate of obesity in males of age 35–39 which followed by a slight decrease in the rate of obesity in the males of age 40–44. However, the highest rate of obesity was observed among the male in the 50–54 years of age (S7 Fig in S1 File). On the other side, the world prevalence of female obesity had more consistence patterns. For example, in almost all of the included periods, the females in their 60–64 had the highest rate of obesity. Interestingly, the rate of obesity decreased slightly after age of 50 and then reached to its peak value in the age group of 60–64. This issue highlights the presence of a possible age effect for this population (S6 Fig in S1 File). Also, it should be note that the highest rate value for male and female obesity prevalence

was different. For instance, the highest value rate for period of 2010–2014 for male was less than 12% however, the same parameter for the population of females was around 19%. Moreover, pairwise comparison of the age-groups in different periods revealed that in most of the periods, females had higher prevalence rates.

A subgroup analysis of sex groups, in different SDI categories, revealed that the trend of obesity in high SDI countries followed a similar pattern to those of global rates for females (S8 Fig in S1 File). However, in males the pattern was slightly different; leaving the age group of 50–54 to had the highest rate without any considerable fluctuations in other age groups (S9 Fig in S1 File). Nonetheless, the net value of the rates was higher in females living in high SDI counties (20% vs 28%). The trend of obesity in males in low SDI countries varied considerably comparing to the global trends. As an example, there was a sharp decrease in the obesity rate in males after age of 70s (from almost 5% to 3%) in the most recent period, although the highest rate was still observed in the males in age of 50s (S10 Fig in S1 File). The trend of obesity prevalence for each of sex subgroups for different SDI groups provided in S8–S17 in S1 File.

Apart from obesity, the prevalence of overweight has also increased. This rising in the prevalence of overweight has been associated with age. Similar to obesity, in the recent periods the rate of overweight was higher and the age distribution of overweight has changed slightly in different periods. The highest rate of overweight was observed in population of people of 50–54 years of age almost in all periods however in the period of 2010–2014, more than 45% of this age-group was overweight which was the highest value comparing to all other age groups in all periods. A similar value for this age group in 1980–1984 was around 35%. In period of 1980–1984, the age group of 70–74 had the highest rate, however in the recent periods a shift toward the younger age groups (Fig 2). In supplementary materials we provided a full description of overweight trend in different sex and SDI groups (S18–S34 Figs in S1 File).

After describing different trends of obesity and overweight based on the three-time varying parameters, we compared the different possible model estimations based on these three elements. Usually, in majority of the available tools for APC analysis, the logarithmic link function (a Poisson model) used to calculate the coefficients of the each of the three parameters. Table 1 shows the deviance table for the global prevalence of obesity. Based on AIC value (239390.2) the APC model was the preferred model to be selected. However, due to the presence of overdispersion in our Poisson model, a quasi-Poisson model was implemented rather than a Poisson model.

Based on our analysis, the age had inverted U shaper effect on the trend of obesity in the world (Fig 3g). The highest coefficient is for the age group of 50−54 (detrended $\alpha_{50} = 1.02$, P-value <0.001). In term of period effect the peak effect was observe for period of 2005−2009 (detrended $\beta_{2005} = 0.022$, P-value <0.001) while the lowest period effect belonged to the period of 1990−1994 (detrended $\beta_{1995} = -0.057$, P-value <0.001) (Fig 3h). The cohort effect had negative impact on the trend of obesity in the world; the highest impact was observed for the cohorts of 1945 (Fig 3i). It should be noted that these coefficients were normalized to be zero at the two tails of the parameters value.

Table 2 presents the deviance table for the global overweight prevalence. Similar to obesity, AIC value for the APC model was the lowest possible value (515525.9) and none of the other sub-models did better than APC. Therefore, the APC model was used to calculate the effect of three time-related parameter on prevalence of overweight. However, since the overdispersion existed here, we choose a quasi-Poisson model to compute the time effects on the prevalence of overweight.

In the Fig 4 the normalized effect of age, period, and cohort can be observable. Like obesity, the age had an inverted U-shaped effect on the prevalence of overweight. Similarly, the age group of 50–54 had the highest effect (detrended $\alpha_{50–54} = 0.84$, P-value <0.001). Besides the age of 45–49 had also a same effect of the trend of overweight in the world (detrended $\alpha_{45–49} = 0.82$, P-value <0.001). In terms of period effect the period of 2000–2005 had highest effect of the overweight trend (detrended $\beta_{2000–2004} = 0.0293$, P-value <0.001). Furthermore, the cohort effect might play role on the trend of overweight in the world. Several cohort groups including cohorts of 1935, 1940, 1945 (strongest effect), 1950, 1975, and 1980 had effect on trend of overweight. The effects among these cohort groups had negative values indicating being born during these years reduced the rate of overweight.

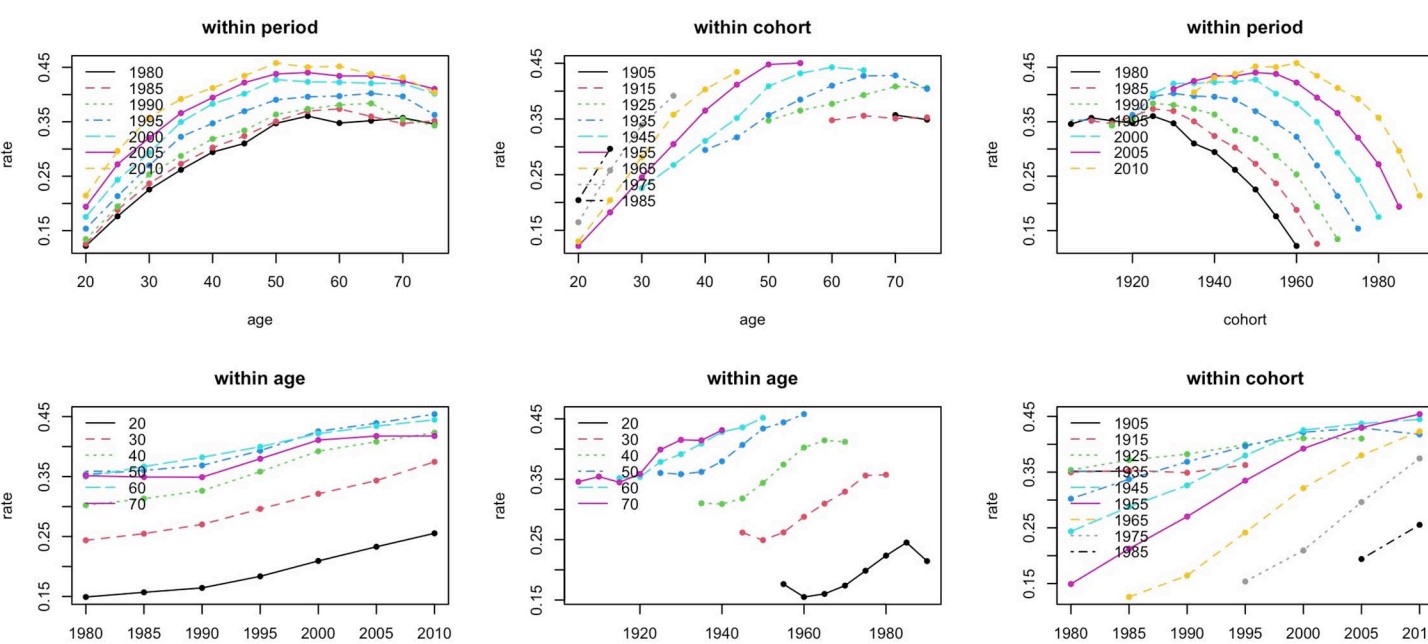

**The pattern of overweight prevalence trend in both genders in the world**

**Fig 2. The chart in the top-left corner indicates that the age distribution of overweight individuals has largely remained consistent over time, irrespective of the period.** In the **top-middle chart**, the overweight prevalence trends within different cohorts as they get older through the study course. The **top-right chart** presents the trends in overweight prevalence for various cohorts within different periods. Here, a noticeable shift in the starting point of each trend line can be observed as time progresses. However, the overall shape of the trend lines remains largely unchanged across the different periods. The **bottom-left chart** depicts how the trends for each age group progress with increasing periods. It demonstrates that every age group shows a higher prevalence of overweight individuals compared to previous periods except for the age group of 70s in period of 1990 which showed a slight decline comparing to the previous period. In the **bottom-middle chart**, the trends for the same age groups across different cohorts are compared. For example, individuals aged 40s who were born around 1940 exhibit lower prevalence rates compared to those in the same age bracket born in 1970. Lastly, the **bottom-right chart** highlights how overweight prevalence trends have evolved for each cohort over time. It indicates that, similar to obesity trends, most cohorts, excluding those from 1905 and 1915, have experienced an increase in rates over subsequent periods.

**Table 1. Comparison of different models to Age-Period-Cohort model as reference for obesity prevalence in the world.** A Poisson model was fitted to data.

| Model | Deviance | DF of residual | LR vs APC | DF vs APC | P-value | AIC |
|-------|----------|----------------|-----------|-----------|---------|-----|
| APC | 237746.49 | 50 | NA | NA | NA | 239390.2 |
| AP | 6030780.8 | 66 | 5793034.35 | 16 | 0.001 | 6032393 |
| AC | 1199057.9 | 55 | 961311.357 | 5 | 0.001 | 1200692 |
| PC | 128125028 | 60 | 127887281 | 10 | 0.001 | 1.28E+08 |
| A | 367541095 | 72 | 367303349 | 22 | 0.001 | 3.68E+08 |
| P | 289632283 | 77 | 289394536 | 27 | 0.001 | 2.9E+08 |
| C | 277924644 | 66 | 277686898 | 16 | 0.001 | 2.78E+08 |

AIC: Akaike Information Criterion, LR statistics: likelihood ratio statistics, DF; degree of freedom

**Fitting a quasi-Poisson model to the prevalence of obesity in both genders in the world**

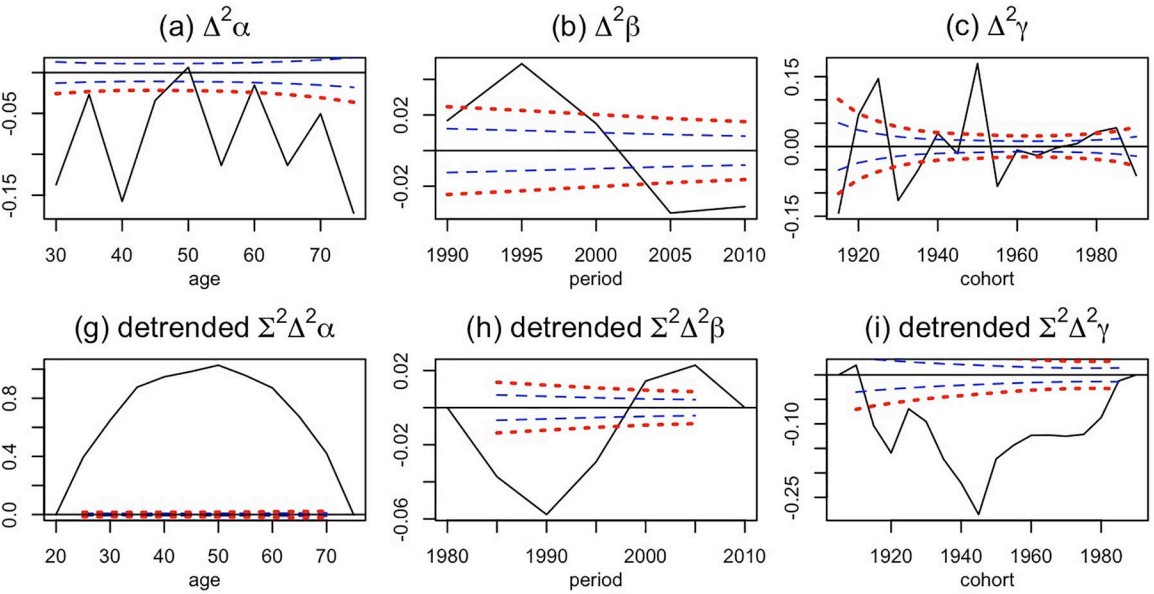

**Fig 3. Plots of the results of the APC model fit to global obesity prevalence.** The blue dashed and red dotted lines are one and two standard errors around the zero, respectively. The plots in the top row show the canonical coefficients of each time-varying parameter. These coefficients explained the effect of the double difference of each effect on the trend, which was further translated into the acceleration or growth rate of the effect. On the other hand, the bottom row showed the normalized effect. The normalization was done on the basis of the first and the last value of the parameters.

**Table 2. Comparison of different models to Age-Period-Cohort model as reference for overweight prevalence in the world. A Poisson model was fitted to data.**

| Model | Deviance | DF of residual | LR vs APC | DF vs APC | P-value | AIC |
|---|---|---|---|---|---|---|
| APC | 513775.97 | 50 | NA | NA | NA | 515525.9 |
| AP | 16700884 | 66 | 16187107.7 | 16 | 0.001 | 16702602 |
| AC | 2188362.9 | 55 | 1674586.96 | 5 | 0.001 | 2190103 |
| PC | 320300540.8 | 60 | 319786764.8 | 10 | 0.001 | 3.20E+08 |
| A | 847574529 | 72 | 847060753 | 22 | 0.001 | 8.48E+08 |
| P | 1.115E+09 | 77 | 1114181625 | 27 | 0.001 | 1.11E+09 |
| C | 528636812 | 66 | 528123036 | 16 | 0.001 | 5.29E+08 |

AIC: Akaike Information Criterion, LR: likelihood ratio, DF; degree of freedom

The results of our subgroup analysis for both gender and SDI subgroups were provided in supplementary materials. Briefly, in the global scale, the patterns of each of age, period, and cohort effects on trend of obesity were similar to those observed in whole global population (S35 and S36 Figs in S1 File). In high SDI countries, consistently comparable patterns were observed regardless of the sex subgroups (S37–S39 Figs in S1 File). However, in low SDI countries, the effects had different patterns for period and cohort effect. Unlike the high SDI countries, in the low SDI countries cohort had not much impact of the prevalence of obesity, instead a fully negative period effect was observed for trend of obesity in these countries (S49–S51 Figs in S1 File). The results of our subgroup analyses for overweight can be found in S52–S68 Figs in S1 File.

**Fitting a quasi-Poisson model to the global overweight prevalence in both genders**

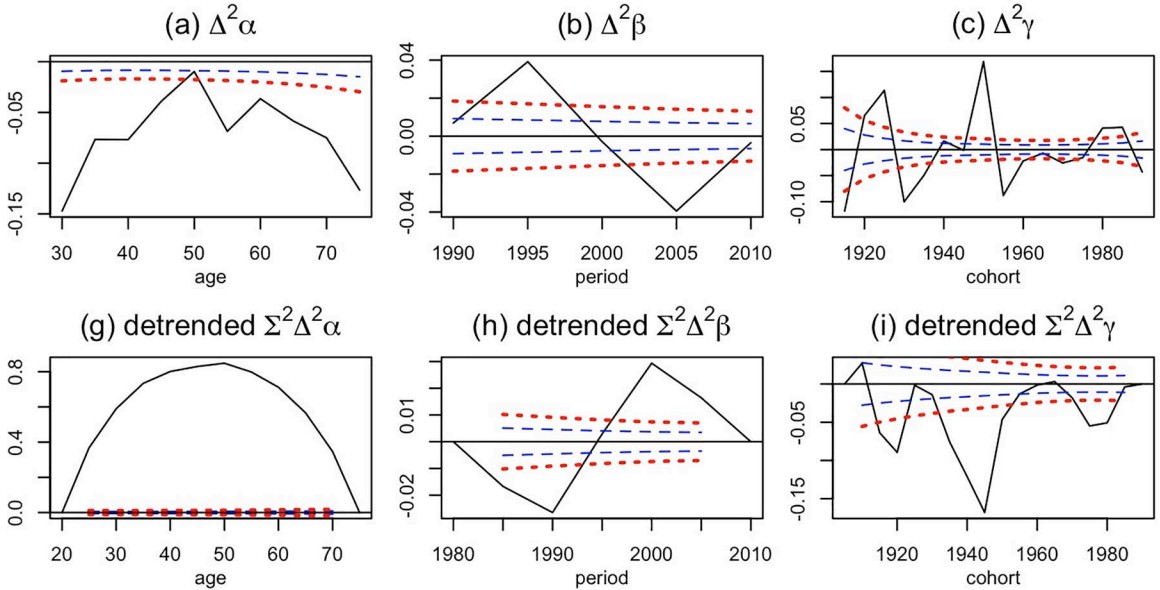

**Fig 4. Plots of the results of the APC model fit on global overweight prevalence.** The plots A to C clarified how the double difference of each effect influenced the trend. Conversely, the bottom row displayed the normalized effect of each time varying parameter, achieved by standardizing the values based on the initial and final parameters.

Our sensitivity analysis using Gaussian link function while considering log normal distribution for the data, showed a slight variation between the current over-dispersed Poisson model and the Gaussian model based on logarithm of the data. Therefore we did not observed a significant change or drastic variation between the outputs of the two model.

## Discussion

This study showed increasing trend in obesity and overweight in the worldwide. However, unlike other similar studies, in this report an ACP model was implemented to understand changes in the prevalence of obesity in a global scale. Our study shows that almost in all periods, the people in the age of 50s had the highest rate of obese and overweight comparing to other age groups. After this group, age of 40s and 60s had the highest rate of obese people. This indicates that the trend of obesity reached to its peak by in the Middle Ages however after this age the rate of obese population started to decrease.

The increasing prevalence of obesity in recent decades is known as a global epidemic [30]. Finkelstein et al. suggested that the prevalence of obesity will continue to increase rapidly, with a 30% rise in prevalence by 2030 [31]. Obesity is a complex interaction between genetic susceptibility and environmental factors [32]. Recent decades have been marked by changes in the food production, storage and processing and consumption of processed foods and ready meals with high calorie and low cost that are more available, made person obese [33].

According to the findings from the Global Burden of Disease (GBD) study in 2015, approximately 603.7 million adults were affected by obesity that year, translating to a prevalence rate of 12% among the global adult population [2]. The data also revealed that, across all levels of Socio-Demographic Index (SDI) and among different age groups, obesity was more common in women than in men. Notably, the highest prevalence was observed in women aged 60–64 living in countries with a high SDI [2]. It is important to highlight, however, that the rates of increase in obesity from 1980 to 2015 were

similar for both genders across all age categories, with the most significant growth occurring in early adulthood for both men and women [2].

On the other hand, in this study we used very different approach to demonstrate and analyze the same data. In this study demonstrate the trend of obesity and overweight prevalence in within different age and period groups. This method of demonstration at frits place provides a different perspective on the data and gave a very clear view on the potential susceptible groups. This study aims to shed light on the potential impact of period effects on obesity and overweight rates across different age groups. In addition, unlike the first study, we examined the influence of cohort or generational effects on the prevalence of obesity and overweight. Although we did not find a significant cohort effect on obesity rates at the global level, we did find that the global prevalence of overweight is indeed affected by generational differences. However, it should be noted that drawing a broad and general picture based only on global rates may be biased because the use of global rates may mask the effect of possible confounders such as economic, social, and medical parameters. Therefore, in this study, we performed APC analysis based on SDI categories.

Based on our findings, it was observed that the effect of period as a time varying parameters, changed during our study period. While it had positive impact on the trend in middle periods for the global trend of obesity, the period effect in low SDI countries had opposite trend. However, it the recent periods the period effect tends to gain increasing effects however it failed to achieve such an impact even in the latest period. On the other side, in the high SDI countries the period effect showed positive impact on the prevalence of obesity through after the period of 2000–2004. More importantly, in other SDI categories such high-middle SDI, initially the period effect had negative effect on the prevalence of obesity but in recent periods it possessed positive effect indicating the presence of factors in the recent periods which that pose positive impact on the obesity prevalence.

A key factor in this regard may be the development of new venues for behavior change, such as the adaptation and increasing implementation of social media in modern lifestyles. Previous research on the possible role of social media in the development of the obesity epidemic mainly emphasizes the positive effect of these new platforms on increasing the prevalence of obesity and overweight [34]. Boyland EJ et al. found that digital food marketing on social media is unescapable and associated with unhealthy food choices in children, but not in adults [35]. In addition, Aghasi M et al. showed in a dose-response meta-analysis that an additional hour of Internet use per day could increase the risk of being overweight or obese by 8% [36]. However, later studies showed that physical, behavioral, and dietary interventions delivered through social media can make a big impact on reducing overweight and obesity in the general population [37]. Furthermore, Loh Y. L et al. demonstrated that using social media-based interventions may be an appropriate solution for helping adults lose weight [38]. This issue is crucial to be considered as policy makers can rely on social media as a tool to counteract the rising trend of obesity and overweight with application of the underlying reason of the rising trend.

Consistently, An R and colleagues employed a fixed-effects APC model to investigate how age, period, and cohort factors affected the prevalence of obesity in the United States [20]. Similar to our study they compute an inverted U-shaped age effect on the pattern of obesity prevalence. Besides, they showed a positive period effect on the rate obesity in United States. Inconsistent to our findings, they failed to achieve any association cohort effect and prevalence of obesity. Schramm S et al. assessed the trend of obesity in Denmark using an APC model. They observed an inverted U-shaped relationship between age and obesity, where the incidence of obesity rose with age until reaching late middle age, after which it declined in older age groups. Additionally, there was a pronounced period effect, indicated by a rising trend in obesity rates over time, while there was no indication of cohort effects [21].

It should be noted that no cohort effect has been observed for prevalence of obesity in considerable number of similar studies. However, in line with our observation, Dobson A et al. found a positive cohort effect for trend of obesity in Australian women [39]. This issue may highlight the fact that aside from technical and methodological issues introduced by some studies to explain this discrepancy, the factors such as economic and social issues may have role in obesity trend. Besides, one possible explanation for the discrepancies between our observation of cohort effect and the current literature

may be the fact that we used global data rather than a country-level dataset. Since the cohort effect was not observed in all subgroup analyses based on SDI categories, this issue highlights the hypothesis that the cohort effect may not be a key factor in determining obesity trends in all geographic locations.

Reducing barriers to trade and global commodities flow expands the Western diet, associated with increased consumption of sugar and fat and reduced consumption of legumes and vegetables, which is a factor in the rising prevalence of obesity in developed and developing countries [40]. Low physical activity and sedentary lifestyles are risk factors of obesity and overweight [41]. Growth of population size, population aging and urbanization especially in developing countries are the factors contributing epidemic of obesity [42]. Changes in people's behavior and the increased number and marketing of snacks have considerable impact on weight gain and incident of obesity [43].

Obesity is a known predisposing factor for many chronic diseases, such as hypertension, different cancers, type 2 diabetes mellitus, and even psychological disorders [44]. Our study shows the prevalence of younger adult obesity increased over the time. Younger adult obesity is linked with an increased risk of complications and one of the major concerns is that this population become obese adult with all the risks and complications [45]. Severe obesity in younger age predicts development of obesity in adulthood and associated with more cardiovascular morbidity and higher risk of all-cause mortality [46].

Similar to the rising rates of obesity, the incidence of being overweight has increased with age in recent decades. This trend has resulted in a higher number of individuals falling into the category of being obese. We found that the rates of being overweight showed greater inflation in the younger cohorts than in the older ones. In general, our study shows an increase in the percentage of overweight and obese people with age and time.

From our study it is apparent that peoples with overweight increased over the generations. Toxic environment reported in 1970 have contribution to increases in the population BMI and prevalence of overweight and in last decades it continued to become more toxic and obesogenic [47]. Spending greater periods of life in obesogenic environment is related to greater risk of overweight of some generations. Later born cohorts are more susceptible to becoming overweight more rapidly meaning that prevention strategies can be effective in children and young adults and can prevent adult obesity.

Several limitations of our study should be noted. Our data is modeled and for some age groups, especially for older ages our data failed to achieve precise estimation of parameters. Besides, based on the GBD 2015 core study [2], some countries and some calendar years lacked proper real data, and this issue might cause problem in true effect of each age, period and cohort. One major limitation of the current study, is the time span of our data. The data used in this study gives insight for trends of obesity and overweight for almost 10 years ago, however, it should be noted that there have been limited resources that provide very detailed data on prevalence of obesity and overweight in different age groups and periods. Even, in the current version of GBD studies, there are not subtypes for high BMI, such as obesity and overweight. This issue impedes us from updating our time span and it was somehow compulsory to work with this dataset. Also, since the duration of study period was limited, our predicted model unsuccessful to show considerable robustness, therefore we omitted the prediction model of APC analysis and only the analysis of determining effect of each parameter were provided in this study. However, APC analysis help to understand the prevalence of obesity in the worldwide, but the current study does not identify the causes of obesity and only examines the prevalence of obesity. Ecological fallacy is possible to occur. We were unable to examine countries in overweight, obesity and severe obesity separately. Due to lack of information, we could not study the prevalence of obesity in subgroups by race/ethnicity.

## Conclusion

The analysis obtained from our study provides a better understanding of the effect of age, period, and cohort on the prevalence of overweight and obesity. We observed possible role of cohort effect on the prevalence of overweight which has not been fully addressed in the literature. On the other hand, the cohort effect was observed in some regions. This

necessitates the country level analysis. Changes in social, cultural, and economic landscapes have the potential to influence individuals' lives on a global scale, leading to widespread impacts such as the increasing prevalence of obesity.

## Supporting information

**S1 File. Contains the output of our analysis of the different SDI and sex subgroups.** This file contains plots of trends of obesity and overweight alongside the corresponding APC analysis on the study subgroups. The other file, titled gather-checklist.docx, contains details and points that should be explained based on the study design. Accordingly, the methodological issues and details that should be observed are addressed in this file.
(DOCX)

## Author contributions

**Conceptualization:** Ali Kabir, Arman Karimi Behnagh.

**Data curation:** Arman Karimi Behnagh.

**Formal analysis:** Ali Kabir, Arman Karimi Behnagh.

**Funding acquisition:** Ali Kabir.

**Methodology:** Ali Kabir, Arman Karimi Behnagh.

**Project administration:** Ali Kabir.

**Supervision:** Ali Kabir.

**Writing – original draft:** Arman Karimi Behnagh.

**Writing – review & editing:** Ali Kabir.

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
