## [Decision Letter · Decision Letter 0]

18 Dec 2024

PONE-D-24-42238Prevalence patterns of overweight and obesity in the world:  An Age-Period-Cohort analysisPLOS ONE

Dear Dr.  Kabir,

Thank you for submitting your manuscript to PLOS ONE. After careful consideration, we feel that it has merit but does not fully meet PLOS ONE’s publication criteria as it currently stands. Therefore, we invite you to submit a revised version of the manuscript that addresses the points raised during the review process.

We look forward to receiving your revised manuscript.

Kind regards,

Zhaoqing Du, Ph.D

Academic Editor

PLOS ONE

Journal Requirements:

“This work was supported by National Institute for Medical Research Development (NIMAD) (Grant numbers [971515]). Dr. Ali Kabir has received this research grant.”

Reviewers' comments:

Reviewer's Responses to Questions

**Comments to the Author**

1. Is the manuscript technically sound, and do the data support the conclusions?

Reviewer #1: Yes

Reviewer #2: Yes

Reviewer #3: Yes

Reviewer #4: Yes

Reviewer #5: Yes

Reviewer #6: Partly

Reviewer #7: Yes

2. Has the statistical analysis been performed appropriately and rigorously? 

Reviewer #1: Yes

Reviewer #2: I Don't Know

Reviewer #3: Yes

Reviewer #4: Yes

Reviewer #5: Yes

Reviewer #6: No

Reviewer #7: Yes

3. Have the authors made all data underlying the findings in their manuscript fully available?

Reviewer #1: Yes

Reviewer #2: Yes

Reviewer #3: Yes

Reviewer #4: Yes

Reviewer #5: Yes

Reviewer #6: Yes

Reviewer #7: No

4. Is the manuscript presented in an intelligible fashion and written in standard English?

Reviewer #1: Yes

Reviewer #2: Yes

Reviewer #3: Yes

Reviewer #4: Yes

Reviewer #5: Yes

Reviewer #6: No

Reviewer #7: No

5. Review Comments to the Author

Reviewer #1: This paper shows the global trends of obesity and overweight from 1990 to 2015 in a five-year interval. The manuscript was well written with a sound methodology. The analysis was robust and the limitations with APC were clearly indicated. The findings highlight increased trend of obesity and overweight, and also exhibit decline rates in recent generations (cohort effect) indicating that the modern lifestyle started to change in favor of become healthier. Overall, this is great paper that will contribute to the body of knowledge in this subject area and would enhance policies aimed to reduce Obesity and its related cardiovascular diseases.

However, the authors need to make a few adjustments.

1. The introduction is well written, but it will benefit readers to define obesity and overweight, and how they are related in the second paragraph.

2. What gap is this study addressing? It is crucial to mention in the introduction, especially in the last paragraph.

3. The results section in the abstract should include some statistics.

4. There are few language issues which should be addressed. The authors should proofread the manuscript and address the typographical issues. Some sentences are too long and need to be simplified.

Reviewer #2: This an organized and well written article discussing an important topic using a customized analysis tool

the study findings make sense and it might reflect the efforts in last decades to address obesity and overweight

Reviewer #3: The manuscript describes a technically sound piece of scientific research with data that supports the conclusions. Experiments have been conducted rigorously, with appropriate controls, replication, and sample sizes. The conclusions are drawn appropriately based on the data presented.

Reviewer #4: 1. Introduction

• Strengths:

o The introduction effectively contextualizes the global epidemic of obesity and overweight, connecting it to environmental, lifestyle, and socioeconomic factors.

o It provides a succinct rationale for using the Age-Period-Cohort (APC) model and highlights its relevance to understanding global trends.

o Recent literature and statistics are referenced to substantiate the problem's importance.

• Areas for Improvement:

o The transition from discussing the general problem to justifying the methodological choice (APC analysis) could be smoother. A clearer explanation of how APC analysis uniquely contributes to understanding the trends would enhance clarity.

o The introduction could briefly preview the study's key findings to better engage the reader.

2. Methods and Materials

• Strengths:

o The data source and processing methods are clearly described, ensuring transparency in the analysis.

o The statistical approach, including model selection and goodness-of-fit assessment, is robust and well-documented.

o Challenges in APC analysis, such as collinearity, are acknowledged, with references to established methods for addressing these issues.

• Areas for Improvement:

o The justification for using older datasets (due to classification differences in recent GBD data) could be expanded to address potential biases or limitations this decision introduces.

o A more detailed explanation of the APC parameters (age, period, cohort) with examples specific to obesity trends might make the methods section more accessible to readers unfamiliar with APC models.

o Ethical considerations for the use of secondary data could be more prominently detailed.

3. Results

• Strengths:

o Results are well-organized, with clear distinctions between obesity and overweight trends across different time-related parameters.

o The use of figures and tables to present model comparisons and prevalence trends is effective.

o Subgroup analyses by gender add depth to the findings.

• Areas for Improvement:

o Some results are described in a way that assumes statistical expertise. Including plain language summaries or key takeaways for non-technical readers would enhance comprehension.

o Figures, such as trend graphs, could benefit from more descriptive titles and labels for accessibility.

o A deeper exploration of discrepancies or unexpected trends (e.g., obesity rates increasing after age 80) would strengthen the interpretation.

4. Discussion

• Strengths:

o The discussion ties findings to global public health implications, emphasizing the need for targeted interventions.

o It draws on a broad range of literature to contextualize results within larger epidemiological and social frameworks.

o Limitations of the study are transparently acknowledged, which adds credibility.

• Areas for Improvement:

o Some interpretations, such as the impact of social media on obesity trends, are speculative and would benefit from supporting evidence or references.

o Greater emphasis on actionable recommendations for public health policy, based on the study's findings, would enhance its practical relevance.

o The discussion could better address potential confounding factors or biases inherent in the dataset or methodology.

o Expand on the implications of findings for public health interventions and policies.

5. Conclusion

• Strengths:

o The conclusion effectively summarizes the findings and their implications for global obesity trends.

o It highlights the importance of continued research and intervention.

• Areas for Improvement:

o The conclusion could explicitly state the novel contributions of this study, especially in the context of global APC analysis.

General Feedback

• Writing Quality: The manuscript is generally well-written but would benefit from professional editing to correct minor grammatical errors and improve sentence flow.

• Discussion: Expand on the implications of findings for public health interventions and policies.

• Figures and Tables: These are valuable in presenting complex data but need improved labeling and captions for clarity.

• Abstract: While concise, the abstract could include specific prevalence statistics to immediately convey the study's significance.

Minor Errors

Methods and Materials

• Error: "This rises a considerable collinearity between the parameters..."

Correction: "This raises a considerable collinearity between the parameters..."

Discussion

• Error: "In this report an ACP model was implemented..."

Correction: "In this report, an APC model was implemented..."

• Error: "Unlike other similar studies, in this report an ACP model was implemented to understand changes in the prevalence of obesity in a global scale."

Correction: "Unlike similar studies, this report used an APC model to understand global changes in obesity prevalence."

• Error: "One key factor in this regard can be the development of new venues for behavioral changes, such as social medias."

Correction: "One key factor could be the emergence of new platforms for behavioral changes, such as social media."

General issue

Error: Inconsistent capitalization of "obesity" and "overweight" throughout the manuscript.

• Suggestion: Standardize capitalization to lowercase unless these terms start a sentence.

Reviewer #5: This manuscript is exceptionally well-crafted, presenting interesting and revealing data that significantly contribute to the field. The writing is clear, concise, and engaging, making complex concepts easy to understand. The methodology is robust, and the data are well-analyzed, ensuring credibility and reproducibility. Added tables effectively complement the text, enhancing comprehension. Overall, this is a high-quality manuscript that provides valuable insights and is a pleasure to read.

Reviewer #6: The present study by Kabir and Behnagh applies age-period-cohort (APC) modeling to the analysis of global trends in the prevalence of obesity. Their study draws its data from the Global Burden of Disease Study 2015 (GBD 2015), which collected health data from 67.8 million individuals in 195 countries for over 25 years to assess the effect of overweight and obesity on the burden of various diseases (the GBD study is described in depth the 2017 NEJM article: DOI: 10.1056/NEJMoa1614362 by GBD 2015 Collaborators). Kabir and Behnagh analyze the GBD 2015 data using a modeling approach by Kuand et al. that addresses the so-called *identification problem* of APC modeling, concerning the challenge of disentangling the interrelated contributions of age, period, and cohort effects (see 2008 Biometrika article doi: 10.1093/biomet/asn026). The authors' main conclusion is that there exists globally a cohort effect of declining obesity across cohorts, and some speculations for the causes of this trend are discussed.

Overall, the goal of discerning cohort effects in global data on obesity is highly interesting and worthwhile. However, the applicability of a general-purpose APC method to a vast and heterogenous data-set cited without further modification is, in my view, questionable. It is plausible that the mechanisms for cohort effects differ by country characteristics (such as by socio-demographic index, food production and distribution systems, policy differences, cultural differences, etc.). This possibility of heterogeneity in the mechanism of cohort effects across countries needs to be considered and taken into account, for example by using a multilevel modeling approach. More generally, a mechanism-based approach to APC modeling may be appropriate for data in which disparate cohort effects may be present across countries (see e.g. the review article by Fosse and Winship: https://doi.org/10.1146/annurev-soc-073018-022616 for a discussion of such methods). The prospects and need for such a mechanism-based analysis should be explicitly discussed.

Major revisions of the article are required to address several issues of problem formulation, analysis, and exposition. Below, I discuss these section by section.

Introduction:

There are several prior studies that have conducted APC analyses of obesity, and these should be cited and discussed explicitly for context and comparison of the cohort effects noted here. For example, see the following BMJ article on APC modeling of obesity prevalence data for Australia https://doi.org/10.1186/s12874-020-0904-8, and the various other APC studies cited therein concerning other countries. Any previously identified cohort effects should serve as a point of comparison for the findings of the present study.

Methods - Data:

The structure and characteristics of the of GBD 2015 dataset needs to be described explicitly, with a view to demonstrating that this data is amenable and appropriate for the proposed APC analysis. Such information as the number of individuals, ranges of key properties (age etc.) and collection methods should be presented. In particular, limitations for resolving cohort effects in the GBD data-set (e.g. related to variability in data quality across sites and times) should be made explicit. **The authors should not expect the reader to know anything about the data source, and therefore all relevant details about the source (sample size, heterogeneity, collection methods etc.) should be summarized.**

Methods - Analysis:

The statistical methods need to be better explained.

**All parameters and variables mentioned in the article need to be explicitly defined, and their factual interpretations made explicit. The author should not expect a reader of PLoS ONE to be an expert in this statistical method.** The authors must provide enough details about the formalism so that the method can be understood by a general reader. The formalism appears identical to that presented in the reference by Kuang et al., where precise definitions of parameters may be found. The structure of the APC model is well-explained in that reference using the concept of a design matrix relating inputs to responses. I suggest presenting the key definitions and equations of this formalism, from which vantage the concept the authors discuss such as that of a canonical parameter etc. may be better understood.

The assumptions of the APC modeling approach used in this study should be made explicit, and the plausibility of these assumptions needs to be discussed. For example, a justification for the choice of link function and the assumptions about the distribution of the outcome variables needs to be discussed and validated.

More fundamentally, if countries differ drastically in their demographic structure, socio-demographic index (SDI), health systems, food systems, measurement differences, etc., is it plausible to look for cohort effects in data pooled from such disparate sources? A multilevel APC model, or some other form of analysis that tracks country-specific factors that can influence cohort effects likely needs to be pursued. In this context, the authors could consider using a mechanism-based approach to APC modeling, and reporting results by country, effectively expanding where feasible the country-by-country analysis given in the 2017 NEJM article.

Results:

Many of the figures and tables are lacking essential descriptions, with numerous abbreviations used without definition (for example in the deviance tables 1 and 2, and probability transform plots in figures 5 and 6, etc.). Again, the author should not expect that the reader will know the mathematical definitions and graphical conventions used in their exposition. The article should provide all of this to the reader. Major improvements in this aspect of the exposition are required.

Finally, the findings in the present work should be compared directly to the results presented in the original GBD NEJM paper, which characterizes obesity trends by age and period. The key additional elements considered in the present study are the cohort effects, and the discussion, accordingly, should focus on these. Finally, for context, the authors should compare their findings to the findings of cohort effects in other studies of obesity prevalence (such as in the BMJ article mentioned above).

Minor comments:

Various minor grammatical and usage errors occur throughout the article. I suggest the authors use proofreading software to find and correct these.

Reviewer #7: The author has tried to make the best use of the available data to the study. There are concerns regarding the language and scientific writing. It is required to make the writing better for publication.

6. PLOS authors have the option to publish the peer review history of their article (what does this mean? ). If published, this will include your full peer review and any attached files.

**Do you want your identity to be public for this peer review?** For information about this choice, including consent withdrawal, please see our Privacy Policy .

Reviewer #1: **Yes: ** Musa Jaiteh

Reviewer #2: No

Reviewer #3: **Yes: ** Juliana Aggrey

Reviewer #4: No

Reviewer #5: **Yes: ** Edna Acosta Perez

Reviewer #6: No

Reviewer #7: No

---

## [Author Response · Author response to Decision Letter 1]

13 Mar 2025

Dear Editor and Reviewers,

As corresponding author, I would like to thank the editor and reviewers for their time while reviewing this manuscript. I would re-submit to PLOS One on behalf of my colleague, this original paper entitled: “Prevalence patterns of overweight and obesity in the world: An Age-Period-Cohort analysis”. All authors have read and approved this revised submission of the manuscript. No persons other than the authors listed have contributed significantly to its preparation.

In this version, we have made all of the suggested changes. We have also revised our analysis to include subgroup analyses based on the Socio-Demographic Index (SDI) subgroups. As requested by the reviewers, we have justified our model selection procedures and provided appropriate references and justifications. In addition, the date of the data used for the analyses was questioned by some of the reviewers. Although we agree with the concerns raised regarding the date of the study dataset, it should be noted that we waited until the publication of the latest GBD study on the obesity and overweight prevalence dataset, which was published on February 6, 2025. Unfortunately, this dataset lacks the proper data format we need to run an APC model with its considered considerations and constraints. So, we stick with the old dataset. On the other hand, we included subgroup analysis to better represent the association of other possible confounding factors to our model. All changes are highlighted in blue. I hope that our delay in submitting the revised manuscript does not inconvenience the editorial board and reviewers.

Reviewer #1: This paper shows the global trends of obesity and overweight from 1990 to 2015 in a five-year interval. The manuscript was well written with a sound methodology. The analysis was robust and the limitations with APC were clearly indicated. The findings highlight increased trend of obesity and overweight, and also exhibit decline rates in recent generations (cohort effect) indicating that the modern lifestyle started to change in favor of become healthier. Overall, this is great paper that will contribute to the body of knowledge in this subject area and would enhance policies aimed to reduce Obesity and its related cardiovascular diseases. However, the authors need to make a few adjustments.

1. The introduction is well written, but it will benefit readers to define obesity and overweight, and how they are related in the second paragraph.

Response: Thank you for this valuable comment. We revised the introduction section and added the definition of obesity and overweight to the second paragraph of the introduction.

2. What gap is this study addressing? It is crucial to mention in the introduction, especially in the last paragraph.

Response: Thank you for this crucial comment. We agree with the reviewer regarding the gap of knowledge that should be addressed in the introduction section. In fact, based on this comment and others from the other reviewers we restructured the introduction section, initially by better definition of the each time-varying effects and the possible role of these effect on the obesity and overweight prevalence trend.

3. The results section in the abstract should include some statistics.

Response: Thank you for this comment. We revised the results section and added more descriptive and analytic details to that.

4. There are few language issues which should be addressed. The authors should proofread the manuscript and address the typographical issues. Some sentences are too long and need to be simplified.

Response: Thank you for this comment. To improve the language of the study we proofread the manuscript by a native English speaker and also used ChatGPT and Grammarly for possible grammatical errors.

Reviewer #2: This an organized and well written article discussing an important topic using a customized analysis tool. The study findings make sense and it might reflect the efforts in last decades to address obesity and overweight.

Response: Thank you for your feedback. We appreciate your time and effort for evaluation of our manuscript.

Reviewer #3: The manuscript describes a technically sound piece of scientific research with data that supports the conclusions. Experiments have been conducted rigorously, with appropriate controls, replication, and sample sizes. The conclusions are drawn appropriately based on the data presented.

Response: Thank you for your feedback. We are really grateful for your time and effort for assessment of our manuscript.

Reviewer #4: 1. Introduction

• Strengths:

o The introduction effectively contextualizes the global epidemic of obesity and overweight, connecting it to environmental, lifestyle, and socioeconomic factors.

o It provides a succinct rationale for using the Age-Period-Cohort (APC) model and highlights its relevance to understanding global trends.

o Recent literature and statistics are referenced to substantiate the problem's importance.

Response: Thank you for your time and dedication reviewing our manuscript. It is highly appreciated when positive feedbacks received from a reviewer.

• Areas for Improvement:

o The transition from discussing the general problem to justifying the methodological choice (APC analysis) could be smoother. A clearer explanation of how APC analysis uniquely contributes to understanding the trends would enhance clarity.

Response: Thank you for this critical comment. We do agree with the reviewer point of view. Thus, in order to better define and describe each time effect, we provided the instances and interpretation of each time-varying effect on the prevalence of obesity and overweight. Therefore, the readers can understand the possible mechanisms that can be attributed to each of the age, period and cohort.

o The introduction could briefly preview the study's key findings to better engage the reader.

Response: Thank you for this important comment. To better engage the reader to the aims and importance of this study, we provided some examples of age-period-cohort analyses on prevalence of the obesity (not overweight since the literature lacks sufficient evidence). Therefore, alongside other revisions we made to the introduction the readers can understand the necessities of conducting this study.

2. Methods and Materials

• Strengths:

o The data source and processing methods are clearly described, ensuring transparency in the analysis.

o The statistical approach, including model selection and goodness-of-fit assessment, is robust and well-documented.

o Challenges in APC analysis, such as collinearity, are acknowledged, with references to established methods for addressing these issues.

• Areas for Improvement:

o The justification for using older datasets (due to classification differences in recent GBD data) could be expanded to address potential biases or limitations this decision introduces.

Response: This comment is really challenging issue. We fully accept the point raised by the reviewer. However, it should be noted that for the study which target a global issue, it is necessary to have a dataset which produced by a solid and coherence methodologies. Moreover, the APC analysis needs a longitudinal dataset with particular format regarding age and period subgroups therefore using a GBD dataset justified here from a methodological perspective. Regarding the time span of the study, again we waited until the IHME published it latest dataset on prevalences of obesity and overweight; (https://ghdx.healthdata.org/record/ihme-data/gbd-2021-adult-obesity-overweight-prevalence-1990-2050). However, this dataset lacks proper details for conduction of a APC model since it lacks the crude number of obese and overweight population.

o A more detailed explanation of the APC parameters (age, period, cohort) with examples specific to obesity trends might make the methods section more accessible to readers unfamiliar with APC models.

Response: Thank you for this crucial comment. We provided such descriptions and details in the introduction section.

o Ethical considerations for the use of secondary data could be more prominently detailed.

Response: Ethical considerations is of vital importance in a scientific work. We do agree with the reviewers point. It should be noted that datasets used in this study have been freely accessible from the web. Aside from that the IHME, the organization produced the datasets, has allowed all the researchers to use their data for non-commercial purposes. Please see (https://www.healthdata.org/Data-tools-practices/data-practices/ihme-free-charge-non-commercial-user-agreement). Besides, we submitted the protocol of this study for assessment in ethic committee of National Institute for Medical Research Development (NIMAD) in our country.

3. Results

• Strengths:

o Results are well-organized, with clear distinctions between obesity and overweight trends across different time-related parameters.

o The use of figures and tables to present model comparisons and prevalence trends is effective.

o Subgroup analyses by gender add depth to the findings.

• Areas for Improvement:

o Some results are described in a way that assumes statistical expertise. Including plain language summaries or key takeaways for non-technical readers would enhance comprehension.

Response: Thank you for this comment. This comment actually raised a huge problem we had in this study. Usually, the APC analysis performed using a Poisson model, however, our data lacked the key components of conducting a Poisson model due to presence of overdispersion. Therefore, after a comprehensive search on the available tools, we choose the model introduced in the following paper 10.1093/biomet/asn026 and https://www.nuffield.ox.ac.uk/economics/papers/2017/HarnauNielsen2017apcDP.pdf.

In these studies, the authors used canonical reparameterization and chain-ladder model to identify age, period and cohort effects. However, it should be noted that the method they used was really advance and needs a considerable knowledge in statistics and mathematics. Nonetheless, in this version of the manuscript, we tried to simplify the result section and added proper explanation to increase the comprehensibility of this section.

o Figures, such as trend graphs, could benefit from more descriptive titles and labels for accessibility.

Response: Thank you for this valuable comment. We agree with reviewer and since the graphical illustrations have key role in representing the result of the APC analysis, we provide a full description for each plot of the study.

o A deeper exploration of discrepancies or unexpected trends (e.g., obesity rates increasing after age 80) would strengthen the interpretation.

Response: Thank you for this comment. Unfortunately, we had to revise our data in a particular format. Since two to other reviewers asked un to add subgroup analysis based on the SDI subgroups we changed our primary dataset which included the SDI regions data. In this particular dataset, our maximum age was limited to age of 75-79. Since the data for the older age groups provided in aggregated format and since using such particular format violated our primary assumption for conducting APC analysis (equal age and period intervals) we had no choice other than removing older age groups from our data.

4. Discussion

• Strengths:

o The discussion ties findings to global public health implications, emphasizing the need for targeted interventions.

o It draws on a broad range of literature to contextualize results within larger epidemiological and social frameworks.

o Limitations of the study are transparently acknowledged, which adds credibility.

• Areas for Improvement:

o Some interpretations, such as the impact of social media on obesity trends, are speculative and would benefit from supporting evidence or references.

Response: Thank you for this instructive comment. We acknowledge the reviewer’s point. Therefore, we provide a comprehensive literature review with over the effect of social media and risk of obesity and overweight prevalence alongside effect of Meida-based interventions effect of the reduction of obesity.

o Greater emphasis on actionable recommendations for public health policy, based on the study's findings, would enhance its practical relevance.

Response: Thank you for this crucial comment. We agree with this comment as one of the key messages of this study is to inform the policy makers and health expects regarding the targets and key point that should be acknowledged by these people. Therefore, we revised the discussion in this regard.

o The discussion could better address potential confounding factors or biases inherent in the dataset or methodology.

Response: Thank you for this key comment. In fact, this issue was also raised by another reviewer. Therefore, we added details regarding potential confounding factors or biases inherent in our methodology in the method section of the study.

o Expand on the implications of findings for public health interventions and policies.

Response: Thank you for this comment. As requested in the previous comments (two comments above) we revised the discussion accordingly.

5. Conclusion

• Strengths:

o The conclusion effectively summarizes the findings and their implications for global obesity trends.

o It highlights the importance of continued research and intervention.

• Areas for Improvement:

o The conclusion could explicitly state the novel contributions of this study, especially in the context of global APC analysis.

Response: Thank you for this important comment. We revised the conclusion of the study as requested.

General Feedback

• Writing Quality: The manuscript is generally well-written but would benefit from professional editing to correct minor grammatical errors and improve sentence flow.

Response: The text of the manuscript was proofread by a native speaker scientist. Moreover, we used both ChatGPT and Grammarly to improve the language of the study.

• Discussion: Expand on the implications of findings for public health interventions and policies.

Response: As mentioned earlier we revised the discussion accordingly.

• Figures and Tables: These are valuable in presenting complex data but need improved labeling and captions for clarity.

Response: All the figures and tables were regenerated in more transparent format. Also, the legends of the figure have been revised and much more explanation and details were added to increase the comprehensibility of the plots.

• Abstract: While concise, the abstract could include specific prevalence statistics to immediately convey the study's significance.

Response: Thank you for this comment. We do agree with the reviewers point; however, it should be noted that the descriptive analysis of the dataset used, was already provided in the 10.1056/NEJMoa1614362. Besides, reporting the time-related effects without proper explanation and interpretation can mislead the readers. Therefore, instead of providing the value of the estimates we reported the trend of effects.

Minor Errors

Methods and Materials

• Error: "This rises a considerable collinearity between the parameters..."

Correction: "This raises a considerable collinearity between the parameters..."

Response: Sorry for these grammatical mistakes. The manuscript has been corrected as suggested

Discussion

• Error: "In this report an ACP model was implemented..."

Correction: "In this report, an APC model was implemented..."

Response: Sorry for the typo. The manuscript has been corrected as suggested

• Error: "Unlike other similar studies, in this report an ACP model was implemented to understand changes in the prevalence of obesity in a global scale."

Correction: "Unlike similar studies, this report used an APC model to understand global changes in obesity prevalence."

Response: Sorry for these grammatical mistakes. The manuscript has been corrected as suggested

• Error: "One key factor in this regard can be the development of new venues for behavioral changes, such as social medias."

Correction: "One key factor could be the emergence of new platforms for behavioral changes, such as social media."

Response: Sorry

---

## [Decision Letter · Decision Letter 1]

30 Apr 2025

Prevalence patterns of overweight and obesity in the world:  An Age-Period-Cohort analysis

PONE-D-24-42238R1

Dear Dr. Kabir,

We’re pleased to inform you that your manuscript has been judged scientifically suitable for publication and will be formally accepted for publication once it meets all outstanding technical requirements.

Kind regards,

Zhaoqing Du, Ph.D

Academic Editor

PLOS ONE

Additional Editor Comments (optional):

Reviewers' comments:

Reviewer's Responses to Questions

**Comments to the Author**

1. If the authors have adequately addressed your comments raised in a previous round of review and you feel that this manuscript is now acceptable for publication, you may indicate that here to bypass the “Comments to the Author” section, enter your conflict of interest statement in the “Confidential to Editor” section, and submit your "Accept" recommendation.

Reviewer #1: All comments have been addressed

Reviewer #3: All comments have been addressed

Reviewer #5: All comments have been addressed

2. Is the manuscript technically sound, and do the data support the conclusions?

Reviewer #1: Yes

Reviewer #3: Yes

Reviewer #5: Yes

3. Has the statistical analysis been performed appropriately and rigorously? 

Reviewer #1: Yes

Reviewer #3: Yes

Reviewer #5: Yes

4. Have the authors made all data underlying the findings in their manuscript fully available?

Reviewer #1: Yes

Reviewer #3: Yes

Reviewer #5: Yes

5. Is the manuscript presented in an intelligible fashion and written in standard English?

Reviewer #1: Yes

Reviewer #3: Yes

Reviewer #5: Yes

6. Review Comments to the Author

Reviewer #1: The authors have adequately addressed my comments, and the work reads very well. No additional comments from my end.

Reviewer #3: Critical Review of Revised Manuscript

Title & Abstract

• The title remains relevant and captures the essence of the study. However, ensure clarity in specifying the geographical scope or digitalization aspect in greater detail.

• The abstract summarizes the study well but could benefit from a more explicit articulation of key findings and their implications. Consider refining the conclusion to emphasize how digitalization influences SME tax compliance practically.

Introduction

• The introduction adequately frames the problem statement, but it could still benefit from a more structured link between tax compliance challenges and digitalization.

• The research gap is identified, but the justification for the study needs further emphasis on why digitalization is crucial in the context of SMEs beyond efficiency.

• Citations should be updated where necessary, ensuring recent sources (2020 and above) back key claims.

Literature Review

• The theoretical framework is clear, but a stronger connection between the theories and empirical findings should be made.

• The empirical review sufficiently covers relevant studies, but some sections require more engagement with the literature beyond summarization.

• Ensure consistency in writing style and the use of present tense.

• Some citations need refinement to align with APA referencing style.

Methodology

• The methodology is well detailed, but further justification of the sampling technique would enhance clarity.

• The data collection process is well articulated, but ensuring that ethical considerations are more explicitly stated would strengthen the credibility of the study.

• Consider briefly mentioning the limitations of the methodology here, rather than leaving all discussion of limitations for the conclusion.

Results & Discussion

• The results are presented clearly, but further alignment with the research objectives and hypotheses would improve coherence.

• The discussion section should explicitly compare findings with existing literature to strengthen arguments.

• Some claims require more direct referencing to data presented in the results section.

• Ensure clarity in explaining how digitalization practically impacts SME tax compliance beyond just statistical significance.

Conclusion & Recommendations

• The conclusion effectively summarizes key findings but should reiterate their policy and managerial implications.

• Recommendations should be practical and actionable, particularly addressing how policymakers and SME owners can leverage digitalization for tax compliance improvements.

• Consider adding a brief section on future research directions.

Referencing & Formatting

• Ensure all references follow APA style consistently.

• Some in-text citations need proper formatting (e.g., author-year consistency).

• The overall structure follows academic conventions but could benefit from improved transitions between sections.

Final Thoughts

• The revisions have strengthened the manuscript, but there is room for further clarity and alignment between sections.

• Addressing the highlighted points will enhance the coherence, impact, and academic rigor of the paper.

Reviewer #5: The author has carefully considered and addressed all feedback provided by the reviewer, making the necessary revisions to enhance the quality of the work. The author expresses sincere gratitude for the thorough and constructive reviews, which have greatly contributed to the improvement of the manuscript.

7. PLOS authors have the option to publish the peer review history of their article (what does this mean? ). If published, this will include your full peer review and any attached files.

**Do you want your identity to be public for this peer review?** For information about this choice, including consent withdrawal, please see our Privacy Policy .

Reviewer #1: **Yes: ** Musa Jaiteh

Reviewer #3: **Yes: ** Juliana Aggrey

Reviewer #5: **Yes: ** Edna Acosta-Pérez

---

## [Editor Report · Acceptance letter]

PONE-D-24-42238R1

PLOS ONE

Dear Dr. Kabir,

I'm pleased to inform you that your manuscript has been deemed suitable for publication in PLOS ONE. Congratulations! Your manuscript is now being handed over to our production team.

Kind regards,

on behalf of

Dr. Zhaoqing Du

Academic Editor

PLOS ONE